# Identification, Safety Assessment, and Antimicrobial Characteristics of Cocci Lactic Acid Bacteria Isolated from Traditional Egyptian Dairy Products

**DOI:** 10.3390/foods13193059

**Published:** 2024-09-26

**Authors:** Khaled Elsaadany, Abeer I. M. EL-Sayed, Sameh Awad

**Affiliations:** 1Department of Dairy Science and Technology, Faculty of Agriculture, Alexandria University, Alexandria 21545, Egypt; elsaadany12@yahoo.com; 2Botany and Microbiology Department, Faculty of Science, Damanhour University, Damanhour 22511, Egypt; abeer.elsayed@sci.dmu.edu.eg

**Keywords:** Egyptian dairy products, cocci lactic acid bacteria, *Pediococcus acidilactici*, *Lactococcus lactis* subsp. *lactis*, *Enterococcus faecalis*, *Enterococcus faecium*

## Abstract

The main objective of this study is to isolate and identify lactic acid bacteria (LAB) from various Egyptian dairy products, examine their antibacterial and hemolysis potential, and ensure their safety when used as starter cultures in different dairy industries. Egyptian dairy products are often made without the use of commercial starter cultures, using raw milk and artisanal methods. The most popular traditional dairy products are Laban Rayeb and Zabady, as well as the cheese varieties of Ras, Domiati, and Karish. The microbial communities used for fermentation and the diversity of lactic acid bacteria are the most important factors that can affect the quality of these products. In order to investigate the diversity of cocci lactic acid bacteria in Egyptian dairy products, 70 samples of raw or fermented milk and cheeses were collected from traditional cheese-making factories, local markets, and farmhouses located in the Delta area of Egypt. Following this, the LAB were isolated from the samples. One hundred fifty-seven isolates of Gram-positive, catalase-negative, and cocci bacterial species were identified via rep-PCR, and some isolates were confirmed using pheS and 16S rRNA gene sequencing, as follows: *Streptococcus infantarius* subsp. *infantarius* (three isolates), *Enterococcus hirae* (three isolates), *Enterococcus faecium* (ninety-six isolates), *Enterococcus faecalis* (forty isolates), *Enterococcus durans* (six isolates), *Lactococcus garvieae* (one isolate), *Pediococcus acidilactici* (seven isolates), and *Lactococcus lactis* subsp. *Lactis* (one isolate). These findings validate that five strains have strong antibacterial activity against *Escherichia coli, Salmonella typhimurium*, and *Listeria monocytogenes*, and one hundred thirty-four strains were safe for hemolysis. The five strains were selected as protective cultures, including *Pediococcus acidilactici*, *Lactococcus lactis* subsp. *lactis*, *E. faecalis*, and *E. faecium*.

## 1. Introduction

Lactic acid bacteria (LAB) have been involved in food preparation and processing from the beginning of recorded history. One of the earliest food preservation methods that is known to exist is lactic fermentation. Pharaohs’ tombs have been discovered to contain hieroglyphics that show the process of manufacturing cheese. Archaic manuscripts from Iraq dated 3200 B.C. also mention fermented dairy products [1,2].

Numerous metabolic traits, including acidification activity, proteolytic activity, bacteriocin synthesis, resistance to bacteriophages, and exopolysaccharide formation, are present in the LAB employed in commercial starting cultures. It has been shown that novel LAB strains, also referred to as “wild strains”, can be obtained from various non-dairy and milk product sources [3]. Interestingly, it was discovered that these strains not only possessed a significant capability for producing amino acids but also possessed the ability to generate peculiar flavor components. When investigated and used in practice, this natural biodiversity may present new opportunities [4]. Microorganisms chosen for starting cultures are typically isolated from the native microbiota of traditional foods because they are suited to the food processing environment and impart qualities of texture, flavor, aroma, and appearance. The primary purpose of starter cultures, which are primarily made up of lactic acid bacteria and are used in animal-based food, is to produce lactic acid quickly. This lowers pH, prevents the growth of pathogenic and spoilage microorganisms, and lengthens the shelf life of fermented foods. Food safety is further enhanced by the creation of additional metabolites, such as lactic acid, acetic acid, propionic acid, benzoic acid, hydrogen peroxide, or bacteriocins [5]. The identification, safety assessment, and antimicrobial characteristics of cocci LAB isolated from various sources are critical in understanding their potential applications in food preservation and health benefits. Cocci LAB, primarily belonging to genera such as Lactococcus, *Pediococcus*, and *Enterococcus*, are recognized for their ability to produce lactic acid, which plays a significant role in food fermentation and preservation by lowering pH and inhibiting pathogenic microorganisms [6].

Accurate techniques for identifying and characterizing LAB are highly sought-after for industrial food processing and other applied microbiology sectors. The taxonomy of LAB has been studied using a variety of techniques. The identification process often involves assessing morphological characteristics, biochemical properties, and genetic sequencing, which collectively ensure the accurate classification of these beneficial bacteria [7]. Studies on the fermentation of carbohydrates frequently produce unclear results and, in several cases, are unable to distinguish between species. Whole-cell protein electrophoresis using sodium dodecyl sulphate polyacrylamide gel electrophoresis (SDS-PAGE) is a commonly used molecular technique. It shows a strong association with nucleic acid homology data [8].

There are already several PCR-based methods available for figuring out the genetic connections between different bacterial strains [9]. (GTG)5 primed rep-PCR is a dependable and quick technique for species identification of enterococci, according to Svec et al. [10]. An evaluation of the (GTG)5-PCR for the identification of Enterococcus spp., the utility of phenylalanyl-tRNA synthase (pheS) and RNA polymerase as subunit (rpoA) gene sequences as species identification techniques for cocci strains were assessed by Naser et al. [11,12].

Natural milk products are characteristic of the region in which they are produced, and dairy products from Egypt may be a potential source of LAB strains with significant industrial features and genetic information related to biodiversity [1]. The advantageous and antibacterial characteristics of *Enterococcus* species, especially *Enterococcus faecium* and *Enterococcus faecalis*, are frequently researched, utilizing techniques like disk diffusion or broth microdilution to ascertain the *Enterococcus* isolates’ resistance profiles to common antibiotics (e.g., vancomycin, ampicillin, and gentamicin) [13]. Tests for enterococcus isolates on blood agar are typically performed to look for β-hemolysis, a sign of possible pathogenicity [14].

Isolates of *Enterococcus*, especially those from gastrointestinal and dietary sources, have a great deal of promise for antibacterial uses [15]. To guarantee their safe use in food and health products, proper isolation, identification, and safety assessment are essential. Their ability to produce organic acids, bacteriocin, and competitive exclusion, among other antimicrobial qualities, makes them excellent choices for improving food safety and public health. When pathogenic strains of *Enterococcus* are present, certain strains of the bacteria may be virulent due to their synthesis of hemolysin and gelatinase, among other virulence factors. It is crucial to distinguish between strains that could be advantageous and those that could be detrimental by conducting thorough screening and safety assessments [16].

*Lactococcus* species are among the most significant LAB employed in dairy fermentations, particularly *Lactococcus lactis*. *Lactococcus* species increase the safety and shelf life of dairy products by inhibiting infections and spoilage organisms through the production of bacteriocins (such as nisin) and lactic acid [17]. Compared to *Lactococcus* and *Streptococcus*, *Pediococcus* species are employed less commonly in dairy products, but they are nevertheless essential to several conventional dairy fermentations [18].

Most of the starter cultures used are isolated from particular (traditional) products and/or from the production environments associated with those products. This suggests that the potential antagonistic effect against foodborne and spoilage bacteria, as well as its inhibitory effect, must be evaluated beforehand within real food products and in vitro. Because the type of fermented food, the method used, the ripening circumstances, the raw materials, and other elements will all affect its function, choosing a starting culture should be carried out within the parameters of its intended purpose [5]. In addition to antimicrobial and screening for chemicals that inhibit bacteria, for that hemolysis testing of bacteria must be taken into consideration when evaluating the safety of LAB and probiotics. In order to employ these isolates as a protective culture for food products, this study aimed to isolate and identify Gram-positive, catalase-negative, and cocci-shaped bacteria with safety qualities and strong antibacterial activity against *Escherichia coli, Salmonella typhimurium*, *Staphylococcus aureus*, and *Listeria monocytogenes* due to their significant role in food safety and preservation.

## 2. Materials and Methods

### 2.1. Samples

A total of 70 samples of Egyptian dairy products, including milk (15), fermented milk (25), cheese (28), and salted whey (2), were collected from different areas in the Delta region of Egypt from small dairy farms and cheese factories; each sample represents a geographical region. There were samples of both cow and buffalo milk. In Egypt, Zabady, a food that resembles yogurt, is the most popular fermented milk beverage. Laban Rayeb is another common traditional fermented milk. Samples of soft cheeses included Mish, Labneh, and Domiati. Similar to Greek “Kefalotiri”, Ras cheese is a hard cheese from Egypt. The samples were taken aseptically in sterile cups and stored in an insulated box at 4 °C until they were brought to the lab. Typically, samples were examined the same day they were collected.

### 2.2. Isolation of Lactic Acid Bacteria

To aid in their isolation, three grams of each sample were grown in thirty ml of sterilized skim milk and incubated at 30 °C for mesophilic LAB, 42 °C for thermophilic LAB, and 37 °C for both LAB until coagulation for four hours. The cultures were streaked using M17 medium as it is more suitable for isolation of *Lactococcus and Streptococcus* [19], and they were incubated for 48 h. Morphological and physiological tests were used to pre-identify the isolates. LAB colonies were selected based on their morphology and biochemical characteristics, which included Gram’s reaction, catalase test, spore staining, and oxidase utilization assay. A positive and a negative reference strain have been carried out for each test. Colonies with characteristics of Gram-positive, none spore-forming, catalase, and oxidase utilization negative were considered LAB [20,21]. The pure strains were subcultured in 10% reconstitute skimmed milk that contains 15% glycerol, and 5 copies of each strain were stored at −80 °C for further use [4]. 

### 2.3. Identification and Cluster Analysis

Following multiple transfers and plating on M17 agar, all strains were examined for purity. They were then pre-identified using a restricted set of phenotypic assays, including colony and cell morphology, motility, Gram reaction, catalase activity, and oxidase activity. For the purpose of extracting DNA, a single pure colony of Gram-positive, catalase-negative, and cocci were cultivated overnight on M17 at 37 °C.

#### 2.3.1. DNA Extraction

After harvesting and washing the overnight-grown cells in 1 mL of TE-buffer (1 mM EDTA pH 8.0; 10 mM Tris-HCl pH 8.0), the cell suspension was centrifuged for 5 min at 5000 rpm. The supernatants were removed, and the resulting pellet was frozen for an overnight period at –20 °C to aid in the rupture of the Gram-positive cell wall. Subsequently, the defrosted pellet was again mixed with 300 μL of TES buffer and 70 μL of lysozyme–mutanolysin solution (5 mg lysozyme (28262, SERVA, Zandhoven, Belgium) in TE buffer). This was incubated for 60 min at 37 °C. The procedure’s remaining phases were carried out in accordance with Gevers et al. [22]. The final step was to dissolve the generated DNA pellet in 130 μL of TE-buffer and store it at 4 °C for the night. Subsequently, 1.5 μL of an RNase solution (10 mg RNase (SIGMA) mixed in 1 mL milli-Q water) was added, and the RNA was digested for 10 min at 37 °C and then for 10 min at 65 °C. Spectrophotometric measurements at 260/280/234 nm were used to confirm the DNA samples’ quality. Following that, the DNA was diluted to a usable concentration of 50 ng/μL.

#### 2.3.2. PCR Amplification and Cluster Analysis

Gevers et al. [22] described how to perform a cluster analysis of the rep-PCR profiles using the (GTG)5-PCR (50-GTGGTGGTGGTGGTG-30) oligonucleotide primer. The software BioNumerics 7.6.3 (Applied Maths, Sint-Martens-Latem, Belgium) was used for the numerical cluster analyses. Using the unweighted pair group method with the arithmetic average (UPGMA) clustering algorithm, dendrograms were produced. The Pearson correlation coefficient’s percentage values were used to represent the correlation levels.

#### 2.3.3. PCR and Sequencing

The pheS gene primers were employed in the amplification and sequencing processes (Table 1). The target gene of 34 strains was amplified using the primer combinations pheS-21-F/pheS-22-R. In accordance with Naser et al. [11,12], pheS-21-F/pheS-23-R was carried out. Here, 33.5 µL of sterile MilliQ water, 5.0 µL of 10 × PCR buffer, 5.0 µL of dNTPs (2 mM each), 0.5 µL of each of the primers, 50 mM for the forward primer, 50 mM for the reverse primer, 0.5 µL of AmpliTaq DNA Polymerase (1 U/µL), and 5.0 µL of template DNA (0.01 µg/μL) were used in the PCR procedures. A PCR thermocycler (Applied Biosystems, Carlsbad, CA, USA) was used to carry out the PCR and sequence analysis, as well as the data analysis described by Naser [7,8].

For 16S rRNA sequencing, the universal primer 27F (5′-AGAGTTTGATCATGGCTCAG-3′) was used as the forward primer and 1492R (5′-GGTTACCTTGTTACGACTT-3′) was used as the reverse primer in PCR to amplify the 16S rRNA gene. The amplified PCR products of the microbial gene fragments were purified and sequenced at MACROGEN sequencing firm, Seoul, Korea, using an automated sequencer ABI 3100 equipped with the Big Dye Terminator Kit v. 3.1. Sequencing was carried out using primers 518F (5′CCAGCAGCCGCGGTAATACG3′) and 800R (5′TACCAGGGTATCTAATCC3′). After obtaining the sequences, they were compared with the NCBI database through BLAST searches (http://blast.ncbi.nlm.nih.gov/Blast.cgi (accessed on 16 September 2024)). Only the results with reliability higher than 98% were considered for the results.

### 2.4. Hemolytic Activity

Hemolytic activity was measured using blood agar base 2 (Oxoid) plates containing 5% (*v*/*v*) human blood from EL-Shatby Hospital in Alexandria. For 48 h, the plates were incubated at 30, 37, and 42 °C. α- and β-hemolytic responses were recorded in greenish colonies and the clear zone that surrounded them, respectively. For the experiment, two assays were performed. *Streptococcus pyogenes* MGAS 15,252 was used as a β-hemolytic control strain [23].

### 2.5. Antibiotic Susceptibility

Fortina et al. [24] state that the disc diffusion method was employed to assess antibiotic susceptibility. Twenty-two antibiotic discs (Oxoid, Basingstoke, Hampshire, UK) bearing the following information were tested: ampicillin, chloramphenicol, gentamicin, norfloxacin, nitrofurantoin, fusidic acid, lincomycin, ofloxacin, penicillin G, pristinamycin, rifampycin, tetracycline, vancomycin, streptomycin, nalidixic acid, cefotaxim, clindamycin, erythromycin, oxacillin, tobramycin, kanamycin, and ciprofloxacin. Testing was carried out on each antibiotic three times. The discs were put on M17 agar plates that had previously been inoculated with different test cultures. After the plates were incubated at the ideal temperature for 24 h, the diameters of inhibition were measured.

An efficient chemical was used to stop the development of bacteria, and the diameter of the zones of inhibition around the discs was measured. The outcomes receive ratings of resistant, susceptible, and intermediate, in that order.

### 2.6. Screening of Bacteriocin-like Inhibitory Substance (BLIS)-Producing LAB Strains

#### 2.6.1. Preparation of Cell-Free Supernatants (CFSs)

The strains were cultivated in MRS broth for twenty-four hours at the ideal temperature. After extracting the CFS via centrifugation (9500× *g* for 10 min at 4 °C) to remove the antibacterial effect of organic acids, and the pH of the extract was raised to 7.0 using 6 N NaOH. The inhibitory activity of the hydrogen peroxide was lessened with the application of catalase (1 mg/mL) (Sigma, St. Louis, MO, USA). The samples were heated to 100 °C for ten minutes in order to decrease enzyme activity [25].

#### 2.6.2. Antibacterial Activity

The isolated strains of LAB were evaluated for their inhibitory potentials towards various isolates as *Escherichia coli* ATTC 25922 *Salmonella typhimurium* ATTC 14028, *Listeria monocytogenes* ATTC 7644, and *Staphylococcus aureus* coagulase-positive ATTC 25923 were carried out according to Tejero-Sarinena [26]. Inhibitory zones were observed and detected.

#### 2.6.3. Statistical Analysis

The data relating to the inhibitory zones of LAB were analyzed using SPSS version 20 (SPSS Inc., Chicago, IL, USA) and presented as mean based on three independent experiments and triplicate analyses. One-way ANOVA followed by Tukey’s post hoc test was employed for statistical comparison analysis. Data with *p* < 0.05 were considered to be statistically significant.

## 3. Results and Discussion

### 3.1. Lactic Acid Bacterial Content from Different Dairy Products

The count of lactic acid bacteria on the M17 at 37 °C ranged from log10 3.2 to 8.1, log10 4.6 to 7.9, log10 3.4 to 8.3, and log10 8.2 to 8.8 CFU/mL or g of raw milk, fermented milk “Zabady” soft Domiati cheese, and Ras hard cheese, respectively. Domiati cheese had the highest enterococci count (log10 5.4–8.1 CFU/g), while the Zabady had the lowest (log10 1.7–3.5 CFU/g). The high content of salt (8–14% *w*/*v* NaCl) in Domiati cheese could be the cause of this genus’ dominance. The low count of enterococi in the case of Zabady could be attributed to the intense heat treatment of the raw milk, which destroyed the milk’s native microbial flora. In some earlier publications [1,4,8], Cocci LAB, particularly from the genera *Lactococcus*, *Pediococcus*, and *Enterococcus*, were found to be prominent in dairy environments due to their ability to ferment lactose and produce lactic acid, which is crucial for the preservation of dairy products.

#### Strain Typing via Genotypic Methods

In Egyptian dairy product samples, 157 isolates were identified as cocci LAB belonging to *E. faecium* (96), *E. faecalis* (40), *E. durans* (6), *E. hirae* (3), *L. garvieae* (1), *St. infantarius* subsp. *infantarius* (3), *P. acidilactici* (7), and *L. lactis* subsp. *lactis* (1) using a rep-PCR technique, pheS, or 16S rRNA sequencing. Out of 157 identified isolates, 89 were completely identified using GTG5-rep PCR, while the remaining 68 isolates required pheS or 16S rRNA sequencing, as 60 of them were identified using pheS sequencing. Eight more isolates that were not identified via pheS sequencing were fully identified using 16S rRNA, as summarized in Figure 1.

*The St. infantarius* subsp. *infantarius* strain was isolated from Domiati cheese, cow raw milk, and Rayeb milk, and the *Lc. lactis* subsp. *Lactis* and, *L. garvieae* strains were only isolated from Domiati cheese. From every sample of raw milk, cheese, Kishk, and fermented milk, *E. faecium* was isolated. The only LAB species that could be isolated from Zabady was *E. faecium*. The identified strains isolated from Ras cheese were *E. faecalis*, *E. faecium*, *E. hirae*, and *P. acidilactici*; the strains from Rayeb milk were *E. faecium*, *E. durans*, *St. infantarius* subsp. *Infantarius*, *P. acidilactici*, *E. faecalis*, and *E. hirae*; the strains from Karish cheese were *E. faecium*, *E. faecalis*, and *E. duran*; and the strains from Domiati cheese were *E. faecium*, *E. faecalis*, *L. garvieae*, and *L. lactis* (Table 2 and Figure 1).

Table 2 lists the distribution of distinct LAB species in different dairy products. The distribution of the 157 isolates includes 58 isolates from ripened Ras cheese, 2 from Ras cheese whey, 18 from Domiati cheese, 9 from Karish cheese, 4 from raw buffalo milk, 20 from raw cow milk, 1 from Zabady, 37 from Rayeb, and 8 from Kishk.

Isolating and selecting lactic acid bacteria from their natural habitats is the most effective method for establishing new bacterial cultures for use in science and industry. Some LAB enhance texture and flavor development because of their adaptability in terms of producing a broad range of substance components [27,28]. The dairy industry uses carefully selected starter cultures, which can be single or mixed strains, to reliably produce dairy products of exceptional quality. Therefore, it is always necessary to isolate novel strains with higher natural features to be used for flavor development, texture enhancement, probiotic potential, and antimicrobial production [1,29].

It might also be clear that the most identified strains of lactic acid bacteria from typical dairy products were *Enterococcus* strains, which agree with El Soda et al. [4]. This may also be explained by the remarkable resilience of *Enterococcus* strains to extreme heat treatment, elevated NaCl concentrations, and the low pH found in traditional Egyptian dairy products [4]. With 145 isolates out of 157, *Enterococcus* accounted for the majority of cocci isolates; of these, 96 were classified as *E. faecium*, 40 as *E. faecalis*, 6 as *E. durans*, and 3 as *E. hirae*. 

All of these findings pointed to the *Enterococcus*, *Streptococcus*, *Lactococcus*, and *Pediococcus* genera as a total of 157 cocci isolates were found (Table 2, Figure 1). On the other hand, *Enterococcus* was discovered from the pre-identified isolates that could grow at 45 °C, 10 °C, pH 9.6, and in the presence of 4.5% or 6.5% NaCl. These findings indicate that among all cocci LAB *Enterococcus* spp. are more common in Egyptian dairy products. The findings indicate that while lactobacilli and cocci LAB may be present in traditional Egyptian dairy products under some publications [4,30], this study focused on the cocci LAB. This is in line with earlier research conducted by Ayad et al. [30], who isolated multiple LAB genera from Domiatti cheese, Ras cheese, Mish, Zabady, and Laban Rayeb. These genera included *Lactococcus*, *Lactobacillus*, *Enterococcus*, *Streptococcus*, and *Pediococcus*.

Table 2 shows that the two strains of *L. lactis* subsp. *lactis* and *L. garvieae* were only identified from Domiati cheese. Conversely, *L. lactis* subsp. *lactis* was identified by El-Baradei et al. [31] as the main strain in raw milk cheese. While Ouadghiri et al. [32] found *L. garvieae* in six samples of the analyzed raw milk; El-Baradei et al. [31] also found *L. garvieae* in 100% of the examined raw milk cheese. Only the strains of *L. lactis* subsp. *lactis* and *L. garvieae* were isolated from Domiati cheese; *L. garvieae* was discovered for the first time in Domiati cheese by El-Baradei et al. [31].

Before *St. infantarius* subsp. *infantarius* was initially identified as the predominant species from traditional fermented camel milk in Sudan in 2008 [33], it was less known for its role in dairy fermentation than *Lactococcus lactis*. There are several species in the genus *Streptococcus*, and they range greatly in terms of virulence, safety, applicability, and behavior. Although *St. infantarius* subsp. *infantarius* is classified as a member of the *St. bovis* group, which also contains several pathogen species; it has been discovered to be the predominant species in various commonly consumed and safe African dairy products.

Three strains of isolated cocci LAB were found to be *St. infantarius* subsp. *infantarius* in this investigation; two of the strains were isolated from fermented milk (Rayeb) and one from raw milk. Certain strains of *St. infantarius* isolated from fermented milk products in West Africa have been shown to be harmless by researchers [34]. Due to its association with clinical instances, the safety of *Streptococcus infantarius* subsp. *infantarius* may be questioned. There is some evidence that links *St. infantarius* subsp. *infantarius* to certain clinical conditions [35].

### 3.2. Hemolytic Activity

Blood agar was used to evaluate each isolate in this investigation for the presence of β-hemolysis, a marker of possible pathogenicity. Checking the ability of LAB strains to hemolyze blood on blood agar is one safety test that ensures a safe culture, and non-hemolytic activity is thought to be a safety criterion for selecting a food application culture [36]. Hemolysis was assessed on Columbia blood agar plates containing 5% (*v*/*v*) blood. The plates were then incubated at 37 °C for 48 h. Hemolysis features on blood agar were observed as β-, α-, and γ-hemolysis. Only 23 strains (14.65%) out of 157 strains were able to produce hemolysin, as shown in Table 3. Compared to α-haemolysin producers (5.73%), the majority of positive strains (8.92%) are β-haemolysin producers. *E. faecium* showed 6.25% α-hemolysis and 10.42% β-hemolysis. Overall, 5% of strains of *E. faecalis* exhibit hemolysis activity related to β-hemolysis. *E. durans* is one of the major strains with hemolytic activity, accounting for 16.67% of β-hemolysis. These strains of β-hemolytic enterococci may pose a threat to humans and animals.

Out of 677 strains of lactic acid bacteria, *Enterococcus* spp. (328 strains) were the most prevalent, making enterococci the predominant lactic acid bacteria (56%) in Egyptian dairy products, particularly Domiati cheese [4]. Food safety and the use of strains from the genus *Enterococcus* as starting cultures depend on their safety properties. Therefore, it is crucial to meet the requirements for these strains as excellent acidifiers and aromatic generators, as well as safer tools for use in our foods.

Our results are shown in Table 3, where the majority of the strains (85% of tested strains) showed either no hemolysis or γ-hemolysis. These isolates were considered to be harmless organisms since they were not pathogenic and did not exhibit hemolytic activity [37].

### 3.3. Antibiotic Resistance

Given their antibiotic resistance patterns, the isolates mentioned earlier that do not have both the β-hemolysis characteristic and antibiotic resistance may be potential probiotic strains. Enterococci are currently the third-most prevalent bacterial pathogen linked to nosocomial infections, behind staphylococci and *E. coli*. The main issue is enterococci’s resistance to several widely used antibiotics [38].

The previously mentioned isolates determined as having both the β-hemolysis feature and antibiotic resistance could be prospective probiotic strains based on their antibiotic resistance patterns [39]. The patterns of antibiotic resistance for the 134 isolates with non-hemolytic activity were characterized by considering 22 antibiotics: ampicillin, chloramphenicol, gentamicin, norfloxacin, nitrofurantoin, fusidic acid, lincomycin, ofloxacin, penicillin G, pristinamycin, rifampicin, tetracycline, vancomycin, streptomycin, nalidixic acid, cefotaxime, clindamycin, erythromycin, oxacillin, tobramycin, kanamycin, and ciprofloxacin. The results (Table 4) indicate that most of the 37 *E. faecalis* and 80 *E. faecium* strains exhibited a reasonable level of resistance to ten antibiotics: chloramphenicol, gentamicin, ampicillin, nitrofurantoin, rifampicin, tetracycline, ciprofloxacin, kanamycin, oxacillin, and tobramycin. All isolates were resistant to only five antibiotics (Penicillin G, pristinamycin, vancomycin, streptomycin, and nalidixic acid).

Based on these findings, the majority of the 80 *E. faecium* strains showed a respectable degree of antibiotic sensitivity. Table 4 displays the results as well as the antimicrobial resistance of *E. faecium*. Overall, 22/80 (27.5%) were resistant to rifampicin, which made them the most common type; 16/80 (20%) and 24/80 (30%) were resistant to kanamycin and ciprofloxacin, respectively. The majority of antibiotic-resistant *E. faecalis* isolates, 27/37, were resistant to tetracycline, 25/37 were resistant to gentamicin, 15/37 were resistant to norfloxacin, 15/37 were resistant to rifampicin, and 12/37 were resistant to ciprofloxacin. 

The percentages of isolates resistant to various antibiotics matched with those reported in earlier studies. Malek et al. [13] described 35 native strains of *E. faecium* that were isolated from two Egyptian cheeses. One isolate proved resistant to just two antibiotics (nalixidic acid and streptomycin) out of the thirty-five that were sensitive to twelve different antibiotics. The resistance of the strain varied depending on the use of eight antibiotics. Mannu et al. [40] assessed 40 strains of *E. faecium* isolates from dairy products; 2.5% were resistant to erythromycin, 5% were resistant to levofloxacin, 17.5% were resistant to nitrofurantoin, and 5% were resistant to norfloxacin. While 100% of these isolates were sensitive to penicillin, ampicillin, vancomycin, gentamycin, streptomycin, teicoplanin, and chloramphenicol, 90% of them showed intermediate resistance to the antibiotic erythromycin. While Franciosi et al. [41] demonstrated that although 16 strains of *E. faecalis* and 3 strains of *E. durans* were resistant to kanamycin and tetracycline, they were all susceptible to cephalosporin, chloramphenicol, erythromycin, and vancomycin.

Conversely, only one of the *E. faecalis* and one *E. faecium* isolated showed resistance to cefotaxime, lincomycin, and ofloxacin. Peters et al. [38] identified the species distributions and patterns of antibiotic resistance in enterococci that were isolated from animal-based food in Germany. The microorganisms investigated showed sensitivity to amoxicillin/clavulanic acid and ampicillin. In this study, all the strains examined, including 80 *E. faecalis*, *37 E. faecium*, 4 *E. durans*, 2 *E. hirae*, 2 *St. infantarius* subsp. *infantarius*, 7 *Ped. cidilactici*, 1 *Lac. Garvieae*, and *1 L. lactis* subsp. *Lactis*, were found to be susceptible to five antibiotics: penicillin G, pristinamycin, vancomycin, streptomycin, and nalidixic acid.

Numerous studies that examined cocci LAB isolated from various artisanal dairy samples have demonstrated that antibiotic resistance has different effects in different studies, products, and geographical areas. In order to ascertain the frequency of virulence factors and antibiotic resistance, Valenzuela et al. [42] investigated enterococci isolated from meat, dairy, and vegetable meals in Morocco. Penicillin and gentamicin sensitivity were reported in all 23 *E. faecalis* and 15 *E. faecium* isolates that were tested. Tetracycline resistance was, however, present in a remarkably small proportion of *E. faecium* isolates, just 6.66%. Four strains of *E. faecium* were examined by Favaro et al. [43], who found that they were susceptible to vancomycin but not to penicillin or tetracycline.

### 3.4. Antibacterial Effect of LAB

The antibacterial effect of LAB is associated with the production of various substances with antimicrobial properties, such as organic acids (lactic and acetic acid), hydrogen peroxide, acetaldehyde, diacetyl, carbon dioxide, bacteriocins, and substances that are similar to bacteriocins, according to several studies [44,45]. When examining the antibacterial capabilities of the lactic acid bacteria that were isolated from conventional dairy products, antibacterial activity was demonstrated by a few cocci LAB strains that were isolated and identified from local dairy products and were found to be sensitive to antibiotics. Cell-free supernatants (CFSs) from overnight cultures of isolates were used to achieve this, and their efficacy against *Salmonella typhimurium* and *Escherichia coli*, as well as spore-forming bacteria (*B. cereus*), Gram-positive bacteria (*S. aureus* and *L. monocytogenes*), and Gram-negative bacteria (*B. aureus*), was assessed. The CFSs of strains were neutralized to pH 7 to exclude the antibacterial effect of acidity and treated with catalase to eliminate hydrogen peroxide. This is to screen isolates to determine whether acidity, bacteriocins, and/or H_2_O_2_ contributed to the inhibition effect of the tested strain.

The antibacterial impact of cured, neutralized supernatants that are also treated with catalase (Table 5) shows the diameters of the inhibition zones surrounding wells that were filled with CFSs from the LAB isolates. Cocci LAB’s inhibitory effect seems to vary strain-dependently. As with other foodborne pathogens, technological property testing is crucial to the manufacturing of cheese because it identifies the strains of LAB that can quickly drop the pH during the first step of preparation, which is necessary for coagulation. 

The lactic acid produced could have a direct effect on foodborne pathogen inhibitions. In particular, the antibiotic susceptibility of LAB is one of the safety problems; some *Pediococcus* and *Enterococcus* were examined for antibiotic resistance. When choosing a food application strain, antibiotic resistance is an important consideration because strains with this trait have the potential to spread to the gut microbiome [46]. Out of the 85 isolates that revealed susceptibility to all the antibiotics tested, 25 were randomly selected for antibacterial tests against foodborne pathogens. Table 5 lists *Enterococcus*, *Pediococcus*, and *Lactococcus* among the isolates; the strains with high antimicrobial activities might be selected as promising probiotics for additional research in order to employ them in the production of various artisanal cheeses and fermented dairy products.

The results presented in Table 5 demonstrate whether the wells containing CFSs from the isolates under investigation are surrounded by an inhibitory zone. Out of the twenty-five cocci LAB isolates, it was shown that only seven neutralized CFSs could inhibit *E. coli* from growing: one isolate of each of *Lc. Lactis* subsp. *lactis* and *E. faecalis*, three *E. faecium*, as well as two *P. acidilactici* (Table 5). Furthermore, it was demonstrated that only three strains contained antibiotic compounds against *S. aureus* and *L. monocytogenes*, and only six strains contained antimicrobial compounds against *S. typhimurium.*

The CFSs from one *P. acidilactici* and one *E. faecalis* showed inhibition when it came to Gram-negative indicator bacteria. They were unable to suppress Gram-positive indicator bacteria (*S. aureus* and *L. monocytogenes*). Two additional CFSs from *P. acidilactici* and *Lactococcus lactis* subsp. *lactis* impacted inhibitory activity against both Gram-positive and Gram-negative indicator bacteria. Additionally, Table 5 shows that there was only one strain of *E. faecalis* that inhibited Gram-positive indicator bacteria but not Gram-negative indicator bacteria. According to certain research, LAB isolates that showed an inhibitory effect on foodborne bacteria were present in traditional Egyptian dairy products [30]. The identification, safety assessment, and antimicrobial characteristics of cocci LAB isolated from dairy products are essential for their application in food fermentation and preservation [47]. Safety assessment of *Pediococcus* is critical, particularly in the context of food safety and public health. LAB are generally recognized as safe (GRAS) organisms, which allows for their use in food products without significant health risks [48]. Their safety is attributed to their non-pathogenic nature and the production of antimicrobial compounds that inhibit the growth of spoilage organisms and pathogens. For example, LAB produce organic acids, hydrogen peroxide, and bacteriocins, which contribute to their antimicrobial properties [49]. The antimicrobial characteristics of cocci LAB are particularly noteworthy.

#### Bacteriocin Production and Antagonistic Activity

Three bacterial isolates with potential bacteriocin-genic traits were selected from the strains isolated from dairy product samples. The bacteriocin character of the produced antimicrobial compounds was confirmed based on research previously published by Dos Santos et al. [50] employing proteolytic enzymes (proteinase K, pronase, pepsin, and trypsin), in addition to α-amylase and catalase. After treatment with the relevant proteolytic enzymes, cell-free supernatants from strains of *Lactococcus lactis* subsp. *lactis*, *E. faecalis*, *E. faecium*, and *P. acidilactici* lost their antibacterial properties, indicating that the antimicrobial agent was a protein or polypeptide. Moreover, the strains’ antibacterial activity did not change when α-amylase and catalase were added; this was comparable to the untreated samples, suggesting that neither the antimicrobial agent nor H_2_O_2_ was glycosylated.

Growing in M17 broth at 37 °C, *P. acidilactici* and *Lactococcus lactis* subsp. *lactis* produced a bacteriocin that was effective against *L. monocytogenes*. It is interesting to note that some bacteriocins are active against foodborne illnesses, such as *L. monocytogenes*, but may not hinder the growth of other beneficial strains of LAB, which are used in probiotics and starter cultures. When inhibitory activity was defined as the outcome of an interaction between actively developing cultures of *Pediococcus acidilactici* and *Lactococcus lactis* subsp. *lactis*, then additional strains from the panel of test organisms were suppressed in parallel testing. In addition to inhibiting the growth of *Salmonella typhimurium* and *E. coli*, an actively growing culture of *Pediococcus acidilactici* and *Lactococcus lactis* subsp. *lactis* also reduced the growth of other Gram-positive test bacteria (Table 5).

Previous research indicates that pediocin, produced by *Pediococcus acidilactici* PA003, exhibited a potent antimicrobial effect against *Listeria monocytogenes* and certain other Gram-positive bacteria. This activity persisted at pH 2.0 to 12.0 following a 2 h treatment, as well as at 4 °C and −20 °C for a month and at 40, 60, 80, 100, and 121 °C for a single hour. Furthermore, the addition of proteinase K, pepsin, trypsin, and papain to the cell-free supernatant resulted in either total inactivation or a marked decrease in antibacterial activity. An examination of the adsorption properties of pediocin revealed that pH was a critical factor, while temperature and time had little effect on the adsorption effect of pediocin. According to Wang et al. [51], 98% of the pediocin was adsorbed at pH values close to 6.0, with pH levels below 2.0 or above 10.0 showing the least amount of adsorption. Additionally, bacteriocins produced by LAB, such as those from the *Pediococcus* genus, have shown significant inhibitory activity against foodborne pathogens like *Listeria monocytogenes* and *Staphylococcus aureus* [52]. The ability of LAB to inhibit biofilm formation by pathogens further underscores their potential as natural preservatives in dairy products [53].

## 4. Conclusions

The identification, safety assessment, and antimicrobial characteristics of cocci LAB isolated from dairy products highlight their importance in food processing. The promising selected strains, “*Pediococcus* and *Lactococcus*”, have distinct advantages and are essential to the fermentation of conventional dairy products. Its technological and safety characteristics are considered when selecting a strain for use in functional fermented foods. This study recommended strains for their ability to inhibit foodborne pathogens, which will have a direct impact on enhancing the safety and quality of dairy products. Continued research into the functional properties of these bacteria will enhance their application in the dairy industry and contribute to the development of safer and high-quality fermented products.

## Figures and Tables

**Figure 1 foods-13-03059-f001:**
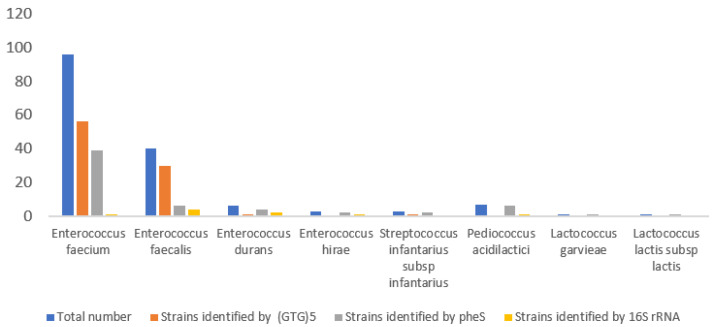
Diversity of cocci lactic acid bacteria identified via (GTG)5-PCR, *PheS* and 16S rRNA sequencing.

**Table 1 foods-13-03059-t001:** Amplification and sequencing primers used in pheS.

Primer Name	Sequence (5′–3′)	Position
pheS-21-F	CAYCCNGCHCGYGAYATGC	557
pheS-22-R	CCWARVCCRAARGCAAARCC	1031
pheS-23-R	GGRTGRACCATVCCNGCHCC	968

**Table 2 foods-13-03059-t002:** Identified strains isolated from Egyptian dairy products.

Dairy Products	Number of Cocci Isolates	Number of Identified Strains	Identified Strains	Method of Confirmation		
				GTG_5_	*pheS*	16S
Buffalo milk	4	4	*Enterococcus faecium*	1	3	
Raw cow milk	20	1	*St. infantarius* subsp. *infantarius*		1	
		13	*Enterococcus faecium*	7	6	
		5	*Enterococcus faecalis*	5		
		1	*Enterococcus durans*		1	
Domiati cheese	19	12	*Enterococcus faecium*	6	5	1
		4	*Enterococcus faecalis*	4		
		1	*Lact garvieae*		1	
		1	*Lc. Lactis* subsp. *lactis*		1	
		1	*St. infantarius* subsp. *infantarius*		1	
Karish cheese	9	4	*Enterococcus faecium*	3	1	
		4	*Enterococcus faecalis*	4		
		1	*Enterococcus durans*		1	
Kishk	8	4	*Enterococcus faecium*	1	3	
		1	*Enterococcus faecalis*	1		
		1	*Enterococcus durans*			1
		2	*Pediococcus acidilactici*		1	1
Ras cheese	58	39	*Enterococcus faecium*	25	14	
		14	*Enterococcus faecalis*	6	5	3
		3	*Pediococcus acidilactici*		3	
		2	Enterococcus hirae		1	1
Rayab	36	1	*St. infantarius* subsp. *infantarius*	1		
		18	*Enterococcus faecium*	12	6	
		12	*Enterococcus faecalis*	11		1
		1	*Pediococcus acidilactici*		1	
		3	*Enterococcus durans*	1	2	
		1	Enterococcus hirae		1	
Ras whey	2	1	*Enterococcus faecium*		1	
		1	*Pediococcus acidilactici*		1	
Zabady	1	1	*Enterococcus faecium*	1		
	157	157		89	60	8

**Table 3 foods-13-03059-t003:** Types of hemolysis of cocci LAB isolated from Egyptian dairy products.

	Total Number	β-Hemolysis	α-Hemolysis	Total	% β-Hemolysis	% α-Hemolysis	%Total
*Enterococcus faecium*	96	10	6	16	10.42	6.25	16.67
*Enterococcus faecalis*	40	2	1	3	5.00	2.50	7.50
*Enterococcus durans*	6	1	1	2	16.67	16.67	33.33
*Enterococcus hirae*	3	1	0	1	33.33	0.00	33.33
*Streptococcus infantarius* subsp. *infantarius*	3	0	1	1	0.00	33.33	33.33
*Pediococcus acidilactici*	7	0	0	0	0.00	0.00	0.00
*Lactococcus garvieae*	1	0	0	0	0.00	0.00	0.00
*Lactococcus lactis* subsp. *lactis*	1	0	0	0	0.00	0.00	0.00
Total	157	14	9	23	8.92	5.73	14.65

**Table 4 foods-13-03059-t004:** Antibiotic susceptibility for cocci LAB isolated from different Egyptian dairy products.

Antibiotic Tests	*En. faecium*		*En. faecalis*		*En. durans*		*En. hirae*		*St. infantarius* Subsp *infantarius*		*Ped. cidilactici*		*Lac. garvieae*		*L. lact.* subsp. *lact.*	
	R	S	R	S	R	S	R	S	R	S	R	S	R	S	R	S
*Number of tested strains*	80		37		4		2		2		7		1		1	
Ampicillin	2	78	4	33	0	4	0	2	0	2	0	7	0	1	0	1
Chloramphenicol	8	72	13	24	0	4	0	2	0	2	0	7	0	1	0	1
Gentamicin	7	73	25	12	0	4	0	2	0	2	0	7	0	1	0	1
Norfloxacin	22	58	15	22	0	4	0	2	0	2	0	7	0	1	0	1
Nitrofurantoin	6	74	3	34	0	4	0	2	0	2	0	7	0	1	0	1
Fusidic acid	2	78	3	34	0	4	0	2	0	2	0	7	0	1	0	1
Lincomycin	1	79	1	36	0	4	0	2	0	2	0	7	0	1	0	1
Ofloxacin	1	79	1	36	0	4	0	2	0	2	0	7	0	1	0	1
Penicillin G	0	80	0	37	0	4	0	2	0	2	0	7	0	1	0	1
Pristinamycin	0	80	0	37	0	4	0	2	0	2	0	7	0	1	0	1
Rifampicin	22	58	15	22	0	4	0	2	0	2	0	7	0	1	0	1
Tetracycline	9	71	27	10	0	4	0	2	0	2	0	7	0	1	0	1
Vancomycin	0	80	0	37	0	4	0	2	0	2	0	7	0	1	0	1
Streptomycin	0	80	0	37	0	4	0	2	0	2	0	7	0	1	0	1
Nalidixic acid	0	80	0	37	0	4	0	2	0	2	0	7	0	1	0	1
Cefotaxime	1	79	1	36	0	4	0	2	0	2	0	7	0	1	0	1
Clindamycin	8	72	2	35	0	4	0	2	0	2	0	7	0	1	0	1
Erythromycin	4	76	1	36	0	4	0	2	0	2	0	7	0	1	0	1
Oxacillin	4	76	3	34	0	4	0	2	0	2	0	7	0	1	0	1
Tobramycin	7	73	8	29	0	4	0	2	0	2	0	7	0	1	0	1
Kanamycin	16	64	7	30	0	4	0	2	0	2	0	7	0	1	0	1
Ciprofloxacin	24	56	12	25	0	4	0	2	0	2	0	7	0	1	0	1

R: microbial resistance; S: microbial sensitivity.

**Table 5 foods-13-03059-t005:** Inhibitory effects (inhibition zone “mm”) of 25 selected Cocci LAB supernatants on Gram-positive and Gram-negative indicator microorganisms.

No		*Source and Code of Isolate*	*E. coli* ATTC 25922			*Salmonella typhimurium* ATTC 14028			*Staphylococcus aureus* ATTC 25923			*Listeria monocytogenes* ATTC 7644		
			Crude	pH 7	Catalase	Crude	pH 7	Catalase	Crude	pH 7	Catalase	Crude	pH 7	Catalase
1	*Pediococcus acidilactici*	Ras-03	22.3 ^a^	18.9 ^b^	17.9 ^b^	17.5 ^b^	13.8 ^d^	12.5 ^d^	9.1 ^g^	5.6 ^n^	7.5 ^l^	8.9 ^h^	7.8 ^l^	4.5 ^m^
2	*Enterococcus faecium*	Raw milk-62	0	0	0	0	0	0	0	0	0	0	0	0
3	*Enterococcus faecium*	Domiati-04	8.5 ^h^	0	0	0	0	0	6.5 ^l^	0	0	0.0	0	0
4	*Enterococcus faecium*	Ras-31	0	0	0	0	0	0	0	0	0	0.0	0	0
5	*Enterococcus faecalis*	Ras-32	0	0	0	12.5 ^d^	0	0	0	0	0	0	0	0
6	*Enterococcus faecium*	Raw milk-60	11.2 ^e^	10.5 ^f^	10.1 ^fg^	4.6 ^n^	0	0	4.5 ^m^	0	0	0	0	0
7	*Enterococcus faecium*	Rayeb-28	0	0	0	0	0	0	0	0	0	0	0	0
8	*Pediococcus acidilactici*	Rayeb-48	0	0	0	0	0	0	0	0	0	0	0	0
9	*Enterococcus faecalis*	Ras-34	22.1 ^a^	0	0	6.8 ^i^	2.3 ^o^	0	21.2 ^a^	0	0	0	0	0
10	*Enterococcus faecalis*	Ras-13	0	0	0	0	0	0	18.5 ^b^	16.5 ^c^	14.5 ^d^	12.6 ^d^	11.6 ^e^	10.5 ^f^
11	*Lact garvieae*	Domiati-81	0	0	0	0	0	0	0	0	0	0	0	0
12	*Lc. Lactis* subsp. *lactis*	Domiati-89	21.5 ^a^	18.6 ^b^	16.1 ^c^	14.5 ^d^	12.5 ^d^	11.9 ^e^	9.2 ^g^	8.9 ^h^	9 ^gh^	5.2 ^l^	4.6 ^l^	4.7 ^lm^
13	*Pediococcus acidilactici*	Whey-20	8.5 ^h^	8.4 ^h^	8.2 ^h^	6.1 ^j^	5.8^k^	4.9 ^l^	0	0	0	4.1 ^m^	3.8 ^n^	3.4 ^n^
14	*Enterococcus faecalis*	Raw milk-76	14.2 ^d^	10 ^fg^	11.6 ^e^	8.5 ^h^	7.9 ^i^	6.8 ^j^	6.1^k^	0	0	5.6 ^kl^	5.2 ^l^	4.9 ^l^
15	*Enterococcus faecalis*	Raw milk-37	4.2 ^m^	0	0	5.1 ^l^	0	0	0	0	0	0	0	0
16	*Enterococcus faecium*	Rayeb-33	0	0	0	4.6 ^m^	0	0	0	0	0	0	0	0
17	*Enterococcus faecium*	Ras-33	12.3 ^e^	10.5 ^f^	11.6 ^e^	10.2 ^f^	7.8 ^h^	7.2 ^ij^	12.5 ^d^	10.5 ^f^	9.8 ^fg^	11.4 ^f^	10.8 ^f^	9.7 ^g^
18	*Enterococcus faecium*	Ras-24	0	0	0	0	0	0	0	0	0	0	0	0
19	*St. infantarius* subsp. *infantarius*	Rayeb-20	8.6 ^h^	0	0	0	0	0	0	0	0	4.3 ^m^	0	0
20	*Enterococcus faecalis*	Ras-07	0	0		0	0	0	0	0	0	0	0	0
21	*Enterococcus faecium*	Rayeb-16	0	0	0	0	0	0	0	0	0	0	0	0
22	*Enterococcus faecium*	Rayeb-53	0	0	0	0	0	0	0	0	0	0	0	0
23	*Enterococcus faecium*	Raw milk-52	9.8 ^g^	2.3 ^o^	0	0	0	0	0	0	0	0	0	0
24	*Enterococcus faecalis*	Rayeb-23	4.5 ^lm^	0		0	0	0	0	0	0	0	0	0
25	*Enterococcus faecium*	Rayeb-21	0	0	0	0	0	0	0	0	0	0	0	0

Means with the different superscript letters are significantly different *(p ≤ 0.05).*

## Data Availability

The original contributions presented in the study are included in the article material, further inquiries can be directed to the corresponding author.

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
