# Peer review of "Identification, Safety Assessment, and Antimicrobial Characteristics of Cocci Lactic Acid Bacteria Isolated from Traditional Egyptian Dairy Products"

_foods, 2024, doi:10.3390/foods13193059_

Round 1
Reviewer 1 Report
Comments and Suggestions for Authors
The manuscript investigated the biodiversity, Safety Assessment, and Antimicrobial Charac teristics of cocci lactic acid bacteria in Egyptian dairy products, which provides valuable information on fermentation, safety and product quality of Egyptian dairy products. However, there are a few corrections and additions that need to be made to improve the quality of the manuscript.
1. Abstract, The research significance of this article must be clearly stated, many key results are unclear or inaccurate, such as 16S rRNA gene sequencing, which is not mentioned in the methodology, “subsp” (Lines 20 and 23) doesn't need italics, “E. coli” should not be abbreviated, “Enterococcus faecalis, Enterococcus faecium (Lines 26 and 27)” should be abbreviated.
2. Introduction Part can be further improved. Why did the author determine the Hemolytic Activity, Antimicrobial Susceptibility, and Screening of Bacterial Like Inhibition Substance (BLIS) Producing LAB strains? What is their relationship with Safety Assessment?
3. Lines 91-99, Please provide a detailed description of the collection of these 70 samples. What are the differences between each type of sample? Is there repeated sampling for each sample?
4. The use of lactic acid bacteria (LAB) and its abbreviation LAB is a bit confusing in the whole manuscript; generally speaking, the abbreviation is uniformly used after the second appearance (Line 45). Please carefully review similar issues in the manuscript and correct them uniformly.
5. Lines 101-105, Please provide a more detailed microbial isolation process. Since 37 °C can separate two types of microorganisms (mesophilic LAB and thermophilic LAB), why did the author set two different temperatures (30 °C and 42 °C) to separate them separately? Why did the author only use one culture medium (M17) in order to isolate as many microbial species as possible? What are the components of this culture medium?
6. Please provide a more detailed information of pre-identification (including colony and cell morphology, motility, Gram reaction, catalase activity, and oxidase activity) and corresponding references.
7. Lines 101, 113, 171, Mixed use of milliliter units, such as milliliter in Line 101, ml in Line 113, 122 and so on, mL in Line 171. Please maintain consistency throughout the entire text.
8. Lines 175-176, The strain number does not need to be italicized, such as ATTC 14028 and ATTC 7644.
9. Lines191-192, “subsp.” does not need to be italicized. Please carefully review similar issues throughout the entire text and correct them.
10. There are many writing errors related to microbial species in the article. Generally speaking, only the species names of microorganisms need to be italicized; When the species name appears for the second time, the genus name needs to be abbreviated (the first letter of the genus name); Please carefully review the entire text and correct similar issues.
11. All tables need to use three line tables according to the specifications.
12. The reference format is not standardized, some journal names are full and italicized (such as International journal of food microbiology in Line 466), and some journal names are abbreviated and not italicized (such as Microbiol Rev in Line 469); Some missed the issue number (such as Reference 6), some missed the page number or article number (such as Reference 12). Please carefully check the entire text and write according to the specifications.
Author Response
Please see the response to the reviewer's comments in attached file

Reviewer 2 Report
Comments and Suggestions for Authors
Dear Authors,
The mauscript entitled: „Identification, Safety Assessment, and Antimicrobial Characteristics of Cocci Lactic Acid Bacteria Isolated from Some Dairy Products” ”, presents an interesting topic. It also has scientific value.
However, the weaknesses of the presented version of the manuscript should also be discussed. The first point is a fact that approximately 50% of the cited publications are very old (in the manuscript file, 26 out of 43 publications date back more than 15 years). Also of great concern is the fact of an outdated research methodology (line 105, reference no. 15). The second point concerns some modifications of a technical nature and serve to clarify the text.
Authors should also consider changing the titles of the presented manuscript to:
„Identification, Safety Assessment, and Antimicrobial Characteristics of Cocci Lactic Acid Bacteria Isolated from Traditional Dairy Egyptian Product”.
However, I have some suggestions for Authors, which are as follows:
• the „Abstract” should be completed with the precise aim of the study: „The objective of this study is …”.
• the section „Introduction” shall be supported by examples of current research of other Authors related to the presented own study,
• please complete a detailed description of the course phenotypic and genotypic characterization of investigated bacterial strains,
• how stock cultures of each bacterial strain were stored for for further studies?
• why in the assessment of the properties of isolated strains of bacteria not used for comparison the well-studied, commercially available, lactic acid bacterial strain?
• line 181: Please provide the appropriate units to express the number of bacteria,
• how many repetitions of the presented study were performed?
• the literature citation throughout the manuscript does not follow the journal's requirements. It needs to be necessarily improved.
• How many repetitions were there for the tests?
• Lack of statistical evaluation methodology,
• Authors should rethink the new strategy for data visualization. It is necessary to enrich the text with figures and necessarily complete with statistical analysis,
• Results need some major revision; please compare your results and give the importance and reasoning in each parameter.
• Please, check the text of the manuscript for grammar mistakes.
• According to the Authors, what is the significance of the results obtained in the context of the trend in biotechnology related to probiotic microorganisms?
• What is the practical relevance of the results obtained by the Authors? What technological significance can isolated strains have?
• This text needs to moderate editing of English language.
• The „Conclusions” should correspond to the aim of the study. This section needs to be improved.
In conclusion, it should be stated that the presented text necessarily requires technical improvement because it does not meet the requirements of the journal for Authors. From my standpoint, this manuscript should be considered for major revision, given the above aspects.
Comments on the Quality of English LanguageIn my opinion, this text needs to moderate editing of English language.
Author Response
Please see the response to the reviewer's comments in the attached file

Round 2
Reviewer 1 Report
Comments and Suggestions for Authors
The author has provided a careful and thorough response to the issues that concern me. I don't have any other comments.
Reviewer 2 Report
Comments and Suggestions for Authors
Dear Authors,
Thank you for response to my review. All previous suggestions has been taking into account. The presented manuscript was improved.